# Exploring lifestyles, work environment and health care experience of Nepalese returnee labour migrants diagnosed with kidney-related problems

**Pramod Regmi**[1]*, **Nirmal Aryal**[1], **Sankalpa Bhattarai**[2], **Arun Sedhain**[3], **Radheshyam Krishna K. C.**[4], **Edwin van Teijlingen**[1]

1 Faculty of Health and Social Sciences, Bournemouth University, Bournemouth, United Kingdom,
2 Research, Policy and Advocacy Unit, Green Tara Nepal, Kathmandu, Nepal, 3 Department of Nephrology, National Academy of Medical Sciences, Kathmandu, Nepal, 4 Migration Health Division, International Organization for Migration, Tripoli, Libya

* pregmi@bournemouth.ac.uk

**Data Availability Statement:** Data is available from Bournemouth University's Research Data Repository. Email: bordar@bournemouth.ac.uk.

## Abstract

In recent years, international media and the scientific community have expressed concerns regarding rising kidney health-related risks among Nepalese labour migrants in Gulf countries and Malaysia. Previous studies have highlighted poor lifestyles and work conditions among Nepalese migrants, which could potentially impact their kidney health. This qualitative study aims to explore the lifestyles and work environment of returnee Nepalese migrants who were diagnosed with kidney health problems. In-depth interviews were carried out with twelve returnee migrants, all males, with half having worked abroad for at least a decade. Our analysis yielded seven themes: (a) living and lifestyles; (b) work environment; (c) exposure to pollutants; (d) Chronic Kidney Disease (CKD) experience; (e) use of painkillers and healthcare; (f) medical expenses for CKD patients; and (g) pre-departure training. This study indicates that Nepalese migrants face numerous challenges, including limited access to clean water and sanitation facilities, poor diets, exposure to occupational hazards, and overuse of pain medication, all of which may contribute to an increased risk of kidney disease. An enhanced pre-departure and on-arrival orientation programme focusing on kidney health-related topics, including the necessary advocacy at the country of destination to provide access to basic services, may encourage migrants to adopt healthy lifestyles and safe working environments, as well as help sensitise migrants to their kidney health risks.

## Introduction

The Department of Foreign Employment of Nepal has issued nearly five million labour permits since 2008/2009. About two million additional labour approvals were renewed since 2011/12 [1]. Lack of employment opportunities, financial insecurity, an unconducive environment for small and medium-scale businesses, and poverty have made Nepalese youth

**Funding:** This work was funded by the COLT Foundation, UK.

**Competing interests:** The authors declare no conflicts of interest. The funder had no role in the design of the study; in the collection, analyses, or interpretation of data; in the writing of the manuscript; or in the decision to publish the results.

desperate to look for jobs abroad. Such desperation, coupled with low education and limited professional skills, aligns with the low-skilled labour shortage in their destination countries in the Gulf and Malaysia [2, 3]. The Government of Nepal has approved more than 100 countries for labour migration, however, six GCC (Gulf Coordination Council) countries and Malaysia are the most popular destinations for Nepalese labour migration as more than 85 per cent of Nepalese migrants (mostly male) work in these countries [1]. There are also many Nepalese migrants working in India, but, due to the open border and lack of a systematic record system, it is hard to estimate the number working there [4]. In recent years, several European countries have also emerged as new destinations for Nepalese migrant workers, taking relatively small numbers each [1].

Labour migration has brought economic and social benefits to the country, the wider society and the families of migrants. This is mirrored in the country's economic indicators, as remittances (2022 estimates) from Nepalese migrant workers have contributed nearly a quarter of the country's Gross Domestic Product (GDP) [5]. However, it comes at a cost. Nearly all migrant workers from Nepal are young adults and from the economically most productive age group of 18 to 44 years (with a median age of 28 years) [1]. The literature has also recorded health and well-being issues of Nepalese labour migrants including poor mental health [6, 7], kidney health problems [8–10], cardiovascular-related deaths and mortality [11–14]. The health vulnerability and risks of Nepalese migrant workers are heightened due to both structural and individual factors such as frequent exposure to occupational, safety and health hazards, poor living/working environments, limited access to health services, and language and cultural barriers [6, 15].

Chronic Kidney Disease (CKD) is the progressive decline in kidney function for over three months [16]. The global prevalence of CKD is around 10% in the general population, but less common in young adults at around 2% [17, 18]. Diabetes and hypertension are the main risk factors for CKD [17]. A nationally representative survey in Nepal reported a CKD prevalence of 4.4% in 20-to-59-year-olds [19]. A systematic review including all published studies estimated the CKD prevalence of 14.6% in the general population in Nepal, but 18.2% in Malaysia, one of the major destination countries of Nepal's migrant workers [20].

Nepal does not currently have a national disease surveillance system, thus data on the health status of returnee migrants is not recorded unless there is a compensation claim for critical illnesses (including kidney failure). Thus, determining whether the considerable number of Nepalese returnee migrants seeking treatment for kidney health problems in hospitals within Nepal reflects a comparable risk within the general population or signifies a disproportionately heightened risk due to occupational and lifestyle-related risks while abroad presents a challenge. Kidney health specialists in South Asia have noted high levels of kidney health-related issues in returnee migrants [8, 21].

There are policies in place in Nepal, the Government requires mandatory provisions such as pre-departure orientation training and health screening for all aspiring migrants. However, no effective measures are in place to identify, record, improve, and support the health and well-being of returnee Nepali migrants. In some destination countries, for example, GCC and Malaysia, post-arrival health check-ups are required [22].

Several studies with Nepalese emigrants have documented poor lifestyles and work environments that may pose a risk to their kidney health. For example, a hospital-based study among 44 returnee migrant workers treated for CKD in Nepal found that 70.5% were involved in manual or semi-manual work in GCC countries and Malaysia, 70% worked more than 60 hours per week [23]. A longitudinal study with 65 Indian construction workers in Saudi Arabia reported that 18% suffered from kidney injury suggesting exposure to heat, long working hours, dehydration, sleep deprivation, and obesity as risk factors [24]. Similarly, a qualitative

study among Indonesian migrant workers with CKD reported unhealthy eating, and a high intake of alcohol and fizzy drinks [25].

A public patient involvement (PPI) workshop with Nepalese migrants and key stakeholders in Malaysia also reported several work-related conditions that might pose a risk to kidney health [9]. For example, factory workers reported a fixed schedule for going to the bathroom or having a drink because the manufacturing process might get disrupted in the absence of workers. Similarly, the unavailability of potable water in many factories, frequent use of high-dose painkillers (often illegally imported so-called herbal painkillers), continuous work for several weeks without a day off, and cheap and possibly hazardous locally brewed alcohol were other issues raised by the workshop participants [9].

The prevalence of CKD in migrant workers is probably underestimated, however, it is clear that their living and working conditions play a crucial role. Therefore, this qualitative study aims to address our knowledge gap in this field, by exploring possible lifestyle and working environment issues experienced by Nepalese migrants diagnosed with kidney health problems.

## Methods and materials

### Study design and participants

This generic qualitative approach study [26] involved in-depth interviews with returnee Nepalese migrants who were diagnosed with kidney disease. Potential interview participants were invited and recruited through collaboration with local female health workers, community-based organizations working for the migrant community, and researchers affiliated with the CKD surveillance programme in the study district. The recruitment of participants and interviews took place from 13th Feb 2023 to 27th March 2023 in the *Dhanusha* District which has the highest number of labour out-migrants in Nepal. The key inclusion criteria were: (i) migrants returned to Nepal in the past three years with at least two years of work experience in GCC countries or Malaysia; and (ii) had a clinical diagnosis of any type of kidney problems. To ensure eligibility researchers accessed and reviewed medical reports from potential participants. Nepal does not have an electronic medical record system and there were no other means to access it. Potential participants were then offered a study information sheet and an agreement form. To capture diverse views, returnee migrants involved in different occupations, working in different, destination countries, and of different age groups were invited and included in the study.

### Interview tool, approach and setting

Our interview guide was based on insights from the literature on kidney health problems in labour migrant populations and, more importantly, insights from Nepalese migrants and key stakeholders through PPI meetings in Malaysia who suggested useful issues to address [9]. Our interview guide included topics such as living and lifestyles abroad, work environment, CKD experience while working abroad, access to health, cost related to kidney health problems treatment abroad, and pre-departure and on-arrival orientation experience.

All interviews were conducted by SB, an experienced male qualitative researcher with a Master's degree in Global Health. PR and NA oriented the researcher about various aspects of qualitative data collection and ethical considerations. The interviews were carried out in the Nepali language at the participant's residence or another mutually agreed location. No other family members of the participants attended the interview. The interviews were audio-recorded with the participant's permission. A concurrent data organisation and analysis approach was followed to enable the interviewer to focus on any emerging issues in subsequent

interviews. Interviews were conducted up to a saturation point where no new information were provided by the participants. The interviewer also made short field notes of non-verbal behaviour during the data collection. Most of the interviews lasted 40–60 minutes.

## Data management and analysis

The data analyses followed the six steps outlined by Braun and Clarke [27], and Microsoft Excel was used to organise, code and track themes in the data. First, the interview recordings were transcribed in Nepali on a Word document and then translated into English by the interviewer. Next, to ensure accuracy, two team members (PRR and NA) of Nepalese origin, independently reviewed and cross-checked the transcriptions with the audio recordings. Each transcript included details about the interview context, nonverbal behaviours, environmental factors, and a reflection on the issues identified in the session. After reading the transcripts PRR independently coded all, while NA and EvT coded half of the transcripts each, serving as second independent coders. Following this, any minor discrepancies in coding were discussed in the team. However, interview transcripts were not returned to the participants for comments, a concept known as member checking, since they were reluctant to read. A thematic approach [27] was followed for data analysis, whereby initial codes were regrouped into subgroups and then into overarching themes. Quotes from the interviews are presented to illustrate the themes. A Consolidated Criteria for Reporting Qualitative Research (COREQ) checklist [28] was followed in reporting the data.

## Ethical consideration

Written informed consent was obtained from all the participants [29]. Through an information sheet in Nepali, the interview participants were provided with detailed information about the study such as study objectives, nature of participant's involvement, voluntary participation, confidentiality, risks and benefits to the participants, and complaint procedure. Personal identifiers were not collected and where necessary information was deidentified to protect participant confidentiality. Ethical approval was obtained from the Nepal Health Research Council (NHRC) and Bournemouth University, UK.

## Findings

### Characteristics of the interview participants

All participants in this study were male returnee migrants, aged 24 to 45. Most (nine out of twelve) were married and had completed at least a secondary education (grade 10). Half of them had spent at least 10 years working abroad (average working duration abroad: 8.9 years, range 4.5 to 15). Some men worked in more than one country. Qatar emerged as the most frequently reported destination on the map among our participants (Fig 1). Four participants (33.3%) were diagnosed with kidney stones, while three (25%) had kidney failure (Table 1).

### Key themes

Seven different themes emerged from our data: (a) living and lifestyles; (b) work environment; (c) exposure to pollutants; (d) CKD experience; (e) use of painkillers and healthcare; (f) medical expenses for CKD patient; and (g) pre-departure training.

**a) Living and lifestyles.** Most participants were in shared accommodations with three to ten fellow migrants. They mentioned having a designated bunk bed within their shared rooms. The following two quotes represent their views on living in a room with other people:

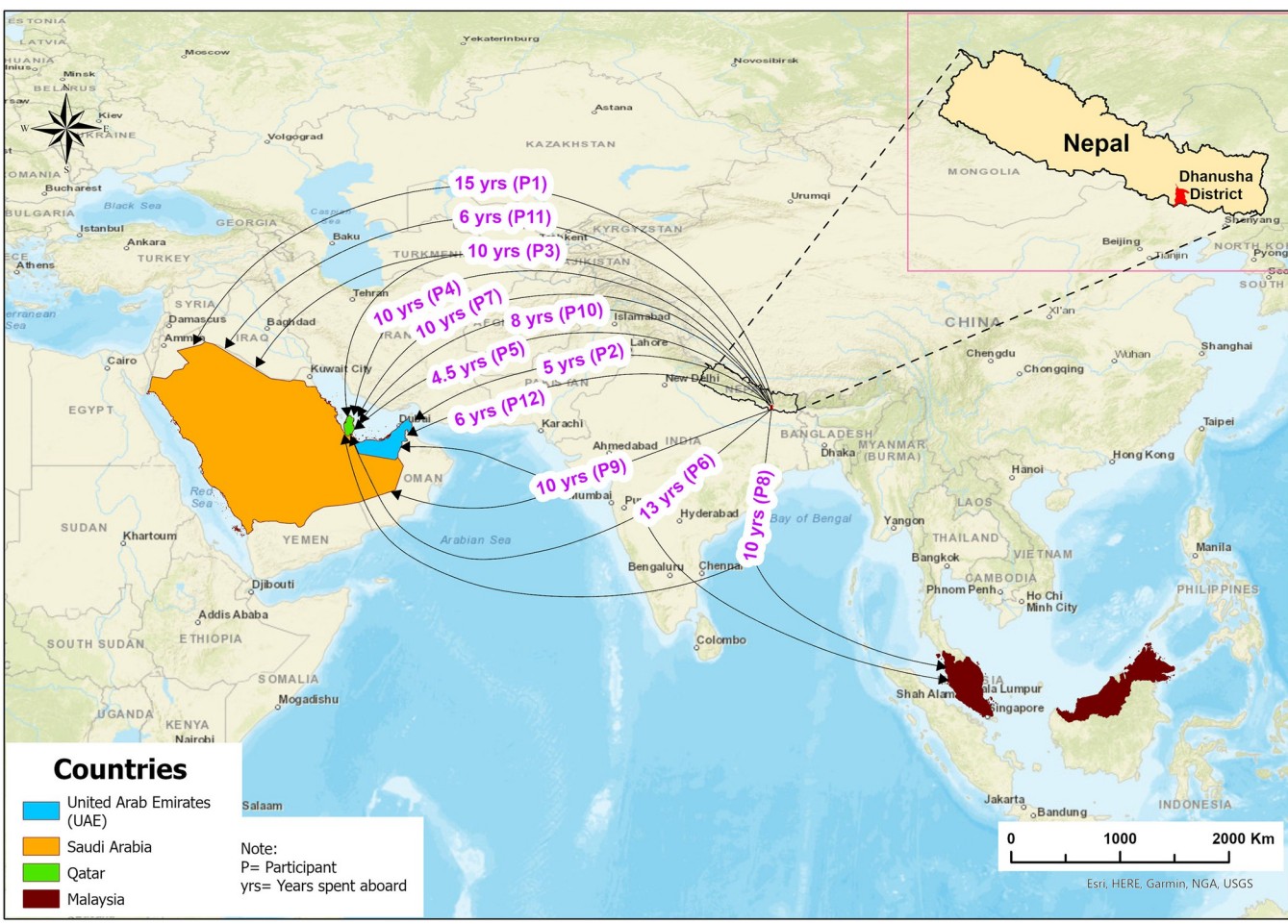

**Fig 1. Destination countries and years worked abroad.**

*"When I was in Qatar, I used to share a room with other 10–12 people. . .the room was given by a company."* (IDI #1)

Some also mentioned problems with facilities, including toilets:

*"So for 10 people there was a single kitchen room and toilet. So, sometime you have to wait for a long queue to use toilet."* (IDI #8)

In response to inquiries about additional amenities like internet access, air conditioning, and drinking water in their accommodation, participants shared that these facilities were supplied by their companies. They further noted the easy availability of drinking water at their accommodations; however, a small number of participants, particularly newcomers, relied on tap water, perhaps for the first few weeks, supplied in their residences before switching to commercial drinking water due to unpleasant taste. Some avoided drinking tap water due to the fear of getting kidney stones, often buying expensive bottled water:

*"While I was in Saudi Arabia for a few months, I drank tap water, it was not good in taste. Later my friends told me if we drank too much tap water it would cause kidney stones and*

**Table 1. Characteristics of interview participants.**

| No | Age (yrs) | Education (highest) | Marital status | Occupation/s | Types of CKD | Types of current treatment |
|---|---|---|---|---|---|---|
| 1 | 45 | 8 | Married | Labourer | End-Stage Renal Disease (ESRD) | Dialysis |
| 2 | 25 | Bachelor's degree | Unmarried | Waiter | Kidney stones | Medication |
| 3 | 33 | 8 | Married | Mason (e.g. plastering) | Kidney stones | Medication |
| 4 | 33 | 12 | Married | AC mechanic | Kidney stones | On medication |
| 5 | 30 | 10 | Married | Waiter | End-Stage Renal Disease (ESRD) | Kidney transplant from India, on medication |
| 6 | 35 | 10 | Married | Cleaner, Electrician, Security guard | Kidney stones | None |
| 7 | 34 | 10 | Married | Barber | Infection both kidneys | Ayurvedic treatment / medicines from India |
| 8 | 24 | 12 | Unmarried | Factory helper Copper plating inboard, Security guard | End-Stage Renal Disease (ESRD) | Kidney transplant from Nepal |
| 9 | 34 | 10 | Married | Mason (brick layer) | Kidney stones | None |
| 10 | 30 | 9 | Married | Cleaner | Kidney stones | None |
| 11 | 32 | 11 | Married | Electrician | Kidney stones | Medication |
| 12 | 28 | 12 | Unmarried | Cleaner | Kidney Stones | Medication |

*they were also not drinking tap water. So eventually I started to drink mineral water [commercially produced water bottles]." (IDI #9)*

A participant who had worked in both Qatar and Malaysia expressed that:

*". . .we didn't need to buy water in Qatar. But, in Malaysia, I had to buy water. I was working as a security guard in a college, so during my duty hours I drank water in the canteen of the college, but we needed to put coins in a water dispenser to buy drinking water while we were in the room." (IDI #6)*

While an interviewee, who had worked in Malaysia, mentioned the need to boil local water, he said:

*"There was a water vending machine. . .outside of our place. We used to buy water from that machine. You have to put shillings paisa [= small coins] in the machine and it will provide good quality water. Many others also drank tap water after boiling it, but I used to buy water." (IDI #8)*

Participants suggested that consuming energy drinks or carbonated beverages gave them energy and that it was widespread among Nepalese migrants:

*"Coke was more preferred, I used to drink one or two litres of coke per day." (IDI #2)*

One person mentioned getting a free sugary carbonated drink every day:

*". . .generally, drinking coke was regular and every day. Our clients used to give us one to two litres of coke daily for free." (IDI #4)*

Most prepared meals in a shared kitchen although few also used their company canteen where they had to pay. There was a practice of preparing excess meals so that they could take some as snacks to work the next day:

*"We used to take leftover food from the dinner and take that to work the next day and have a lunch in between 12 to 1 PM. There was no time to prepare the daytime lunch but at night we used to cook ourselves." (IDI #4)*

Most said that they ate green vegetables regularly, eating fruit was seen as expensive and hence rare, for example:

*". . .I used to eat green vegetables. . .but I rarely ate fruits. I used to eat meat and fish once a week both in the same quantity." (IDI #7)*

Participants reported consuming alcohol frequently, typically on their days off, with some engaging in daily alcohol consumption.

*"During holidays I drank alcohol. There were some Nepalese who used to drink alcohol daily. But, for me I only drank on holidays." (IDI #5)*

*"We used to drink alcohol during our days off. . ..I used to drink 2–3 bottles of beer in one sitting."(IDI #2)*

Regarding the specific types of alcohol consumed, participants mentioned that bottled alcohol and beer were the preferred choices abroad, although a minority opted for locally produced alcohol while in Nepal.

*"I used to drink alcohol while I was abroad. In Nepal, I used to drink alcohol made from millet, but I used to drink packaged (seal packaged) alcohol abroad." (IDI #7)*

Likewise, chewing tobacco was common, most disclosed their longstanding consumption of *Gutka* (a small sachet of tobacco mixed with betel nuts), starting from an early age. Notably, one participant mentioned that consuming *Gutka* contributed to his enhanced mental alertness.

*"For the purpose of remaining mentally active as well, I used to eat about 5–6 Gutka a day while working abroad. But nowadays I consume 2–3 Gutka daily. Before going abroad, I used to consume 6–7 packets per day." (IDI #2)*

**b) Work environment.** Each participant confirmed their daily work often exceeded their standard 8-hour shift. All routinely worked overtime, extending their normal working hours. Additionally, some even worked overtime on their scheduled days off, for example, one participant said:

*". . . I used to work 12 hours daily. . .it depends on you, for how long you will work. Company provided 4–5 holidays in a month, but sometimes I used to do overtime in those holidays." (IDI #3)*

Most reported getting adequate breaks and rest times whilst working long hours. However, one who worked as a security guard in Malaysia reported an absence of facilities for taking rest while on duty:

*"For the first six months, I could not take rest while on guard duty. It was difficult at that time; the temperature was also high. But later after I started to know people and some languages, it was a little easier for me. I found some places to sit and rest." (IDI #6)*

Another participant shared about the working conditions of Nepalese migrants:

*"Situation for Nepalese in the Gulf countries who are working as labourers is very bad. There are no good facilities for food. We were lucky ones who got a good working environment. But for many people living there, the situation is very bad. They don't get enough sleep. The duty hour is sometimes around 12–14 hours. Some had to work under the direct sun for a long time." (IDI #4)*

The majority of our participants were employed as labourers in construction companies or factories. They indicated that the use of personal protective equipment (PPE) was contingent upon the specific regulations set by their respective companies. A participant diagnosed with kidney stones who worked as a mason in a construction company (where airborne cement dust was prevalent) reported using safety equipment while working.

*"We were provided with helmets, goggles, masks for our safety and had to wear them all the time." (IDI #3)*

Another participant employed in a glass factory in Saudi Arabia revealed that wearing a mask was not mandatory in the factory.

*". . .After cutting the glass there used to be a large amount of water, small pieces of glass, and glass dust. It was very difficult to work there. While working we used to wear gloves and helmets only, we didn't wear masks." (IDI #1)*

This perspective was echoed by another working in a factory in Malaysia, where the requirement of wearing a mask depended on proximity to operating machinery:

*"For working near a machine, you need to wear masks. If you are not working near the machine or in other office spaces, you do not have to wear the mask." (IDI #8)*

Most participants reported easy access to drinking water in their workplaces. However, it was noted that in certain companies, a supervisor's approval might be necessary to take a drink break. Many did not have to purchase water at work, as water dispensers were readily available, for example:

*". . .different places inside the office had water dispensers. . .some are even filled with electrolyte solutions, and we could take it when needed." (IDI #3)*

Interestingly, one mentioned that supervisors actively encouraged them to stay hydrated, particularly on hot summer days, by urging them to drink ample water at work.

*"When it was very hot supervisors used to suggest drinking more water." (IDI #6)*

Most indicated unhindered access to toilets despite extended working hours. Nevertheless, a few employed in Malaysia recounted troubling incidents related to restroom access during work hours. For instance, a guard shared his experience as:

*"While working as a security guard in Malaysia, I needed to inform to the supervisor before going to the toilet. I also found that in some companies where Nepalese work they cannot go to drink water and toilet use without the permission of their supervisors. Those Nepalese working*

*as cleaners in tall glass buildings have problems doing toilets when hanging in tall buildings. They carry some kind of plastic with them and urinate there. They told me it is difficult to do such things." (IDI #6)*

One participant, who had undergone kidney transplantation explained, that due to the continuous operation of machinery in the factory (automated function), workers were required to seek permission from supervisors or security personnel to use the restroom during their shifts. This person also revealed that certain companies employed strict security personnel who were reluctant to grant workers permission for restroom or drink breaks, for example:

*". . .A machine in our company ran 24 hours so someone must replace the person going to the toilet, when there was no one to replace them, we used to ask supervisors and seniors for help. In some companies, there are strict bouncers. Workers in those companies often get problems going to the toilet." (IDI #8)*

**c) Exposure to pollutants.** Eight participants disclosed exposure to various pollutants. One participant, who experienced failure of both of his kidneys, had worked 12 years as a labourer in a steel factory in India, and three years in a glass factory in Saudi Arabia, recalled:

*"In India, I used to work in a steel factory. My job was to apply shiny polish to the steel pieces. That polish used to smell. But there I used to wear a mask and all other safety equipment." (IDI #1)*

Another participant, who was employed as a barber and suffered from severe kidney infections, felt there was a link between CKD and having been exposed to various chemicals in the workplace, including lotions, highlighters, straightening creams, fragrances, and cosmetics with unpleasant odours. Despite developing allergies to these substances, he continued working as a barber for an extended period.

*". . .I had to use cosmetic products which generate both fragrances and bad odours. I later developed allergies that caused frequent sneezing and an uneasy sensation while working abroad." (IDI #7)*

A participant with a kidney transplant shared his experiences of working with metal solutions as follows:

*". . .I used to work in a factory that performed copper plating on a printed circuit board (PCB). My job was to dip the PCB into a copper solution. There was a good ventilation system in the factory however, there was always the smell of copper. Masks and safety equipment were compulsory for us." (IDI #8)*

Similarly, another participant working as a cleaner reported his experience as follows:

*"I used to clean floors with detergent. The detergent used to give odour. The floor also gave off an odour after mopping. After a few years I got habituated to that odour." (IDI# 10)*

**d) CKD experience.** None of the participants reported having developed any kidney-related symptoms or problems before going abroad. Among them, seven individuals received a diagnosis of kidney health problems while working overseas, while four participants were

diagnosed after returning to Nepal. Additionally, one participant reported experiencing pain while employed abroad, which was subsequently diagnosed as kidney stones by a doctor in Nepal.

Out of the seven participants diagnosed with kidney disease abroad, three continued their work despite these kidney problems. Conversely, one participant promptly returned to Nepal and commenced dialysis. Another participant, who has now undergone a kidney transplant due to kidney failure while employed in Qatar, continued working for approximately two months before returning to Nepal for treatment:

*"...I became very sick; I had a very high fever and alarmingly high blood pressure of 160/100. Later after staying for four days at the hospital doctors told me that both kidneys had failed. But I worked for about two months because doctors told me that my creatinine level was low and that with medications and a good diet, dialysis could be postponed for a few months. After returning to Nepal, Nepalese doctors in Dharan and Kathmandu also told me the same thing. Thus, I stayed on medication for about a year. After a year my body started to swell, and I started to feel difficulties while urinating. Then after I consulted doctors at Janakpur and started dialysis." (IDI #8)*

Similarly, another participant with severe infections in both kidneys reported working for four months even after his diagnosis. He shared his experience as:

*"After I returned to UAE from Nepal, they did my medical tests, but they did not provide me the reports. Later doctors told me that I had a problem in both of my kidneys. Then, I did a series of tests for about four months, but the exact cause was not determined. At around four months, I did a biopsy of my kidneys which confirmed there was some kind of infection in both of my kidneys. Then doctors gave me some medications, but I never took a rest from my work in those four months." (IDI #7)*

Likewise, someone employed in Qatar recounted being hospitalized due to intense back pain, leading to a diagnosis of kidney stones. After an 11-day hospitalization for treatment, the participant resumed work upon discharge.

**e) Use of painkillers and healthcare.** Participants had different experiences accessing and using medications while working abroad. The majority acknowledged relying on excessive pain relievers, with Paracetamol being the most commonly used. Some also mentioned receiving painkiller injections as prescribed by doctors, although they were unaware of the names or types of the injections.

*"...There were Bengali shops where we could buy Paracetamols or other painkillers. If they did not work, then only we used to go to the hospital." (IDI #3)*

Some were self-medicating when in pain, using readily available over-the-counter drugs:

*"We were like patients ourselves and doctors ourselves, we didn't consult any doctors. We had a day shift and had to work for 10 hours so we didn't have time to visit doctors. When we had body pain, we used to take painkillers like Ibuprofen which we used to bring from Nepal." (IDI #3)*

The majority had taken medications from Nepal. When questioned about the rationale behind this, they expressed a belief that medications abroad might not suit their specific needs.

Additionally, factors such as limited time to consult doctors and the unavailability of certain medicines further motivated them to bring medications from Nepal, for example:

*"I used to ask for cough syrup from Nepal because medicines in Qatar do not fit with my body. . . .Antibiotics are also not easily available there, so we ask for medicines from Nepal and use that as needed." (IDI #7)*

When seeking health care abroad, participants explained that generally companies and hospitals/clinics have agreements between them to provide treatment to workers. The company issues tickets [= referral slips] to the workers who need treatments and also arranges transport if needed. In some cases, referral provisions can be made for specialised care.

*". . .when we are sick, we notify someone from the company living in our quarter, and they then provide us a ticket to go to a clinic. You take the tickets to see doctors and buy medicines." (IDI #5)*

*"After seeing my ultrasound report, the doctor told me that both of my kidneys were damaged. They referred me to another hospital where I was hospitalized for four days." (IDI #8)*

However, getting specialized care from the referral hospital was based on the nature of the employing company. The workers working in mediocre or small-scale companies were returned to Nepal when they felt ill for a prolonged period. One of the participants expressed:

*"Getting healthcare at hospitals depends on the company, my company was good. There was one guy from [= district in Nepal] who felt very ill. The company sent him to the hospital from the site. He had pain in his stomach. After a few days, it was found that he had appendicitis, so he was sent to another hospital for surgery, he joined work after one week. . . so it really depends on the company. If he was employed in a low-scale company, then he would have been handed a ticket to return to Nepal." (IDI #4)*

**f) Medical expenses for kidney health patients.** People held varied opinions on the reimbursement of costs abroad related to kidney treatment. Someone cited a specific instance where the medical expenses for 10–11 days of hospitalization for kidney stone-related pain were eventually covered by his employer.

*". . .We had to take our medical bills to the company and the company gave us money." (IDI #3)*

However, a migrant returning from Qatar mentioned that his company covered only half of the medical costs when his kidneys failed. In contrast, another one's company did not offer any medical compensation at all, just a return ticket to Nepal. Additionally, another participant suffering from kidney infections spent 50,000 Nepalese rupees (equivalent to around USD 375) for kidney treatment and medication while abroad, receiving no reimbursement from the company for the medical expenses.

When questioned about the support received in Nepal, someone presently undergoing dialysis is benefiting from complimentary basic haemodialysis services provided under a scheme to support the poorest patients initiated by the Government of Nepal. Nonetheless, most expressed the challenges in accessing such support in Nepal. Those undergoing dialysis mentioned a weekly cost of 5,000 Nepalese rupees (approximately 37 USD), in addition to other expenses, such as local travel.

One participant with a kidney transplant had not received any assistance from any scheme. His kidney transplant was performed in India, and he continued to consult Indian doctors, rendering his costs ineligible for benefits under this scheme in Nepal.

**g) Pre-departure training.**   Despite the mandatory requirement of pre-departure training for prospective labour migrant workers in Nepal, most participants did not receive any form of pre-departure or post-arrival training. Intriguingly, two participants disclosed that the manpower agency (labour recruitment agent) in Nepal falsified their training certificate without them attending the training.

*". . .I didn't attain pre-departure training, but the manpower company [recruitment agency] provided me with a certificate without any training which I could present in order to take the job."* (IDI #7)

A few with pre-departure training shared that the training primarily offered very basic information such as how to live abroad, where to eat, and what not to do there. They recalled that there was limited coverage on health issues, for example:

*". . .I took that training in Janakpur inside the office of the Chief District Officer. There was not much information about the diseases. However, it provided information on how to live and work abroad."* (IDI #8)

Participants acknowledged the importance of training before going abroad. They agreed that such training may be helpful to promote a healthy lifestyle abroad.

*". . .It would have been better if we had received such training. It would have been helpful if they told us to drink adequate water for our health, live in a safer environment, and protect ourselves from the heat."* (IDI #7)

*". . .it would be better to get information on types of diseases that will occur if we do not drink plenty of water while working abroad, especially in hot places."* (IDI #3)

## Discussion

This qualitative study provides insights into the lifestyles, living and working conditions of returnee Nepalese migrant workers with CKD and other kidney health-related issues as well as their healthcare access experiences while abroad. A unique finding from this study is the migrant workers' pattern of initially use of local tap water and subsequently transitioning to commercially available water, motivated by both the unpleasant taste of tap water, warning from their peers, and their perception of such water causing kidney stones. Switching from tap water to commercial water due to taste issues has been reported elsewhere [30], however, their concerns over their kidney health due to the water taste are less well reported. This suggests further research is necessary with other migrants (e.g., South Asian migrants in the Gulf or Malaysia) on their perceptions of the water they drink.

In consensus with previous literature, our study also documents difficulties faced by Nepalese migrant workers abroad while accessing drinking water and toilet facilities. Similar findings were reported by Nepalese migrants and related stakeholders in a public patient involvement workshop in Malaysia [9]. Most of our participants bought water from markets or vending machines during their stay abroad. It is important to note that most labour migrants come from low socio-economic status, hence paying for drinking water daily not

only adds an extra financial burden to them but also creates barriers to adequate hydration. In addition, most participants reported no issues with shared toilet use in their accommodation, it was however noted that security guards and cleaners working in the tall buildings faced difficulties accessing toilets when needed. This may discourage workers from drinking enough water to avoid the toilet when on duty [9].

Our study found that Nepalese migrant workers often compromise their diet, with a high intake of carbonated/fizzy/energy drinks, alcohol (including counterfeit/homemade alcohol) and smokeless tobacco. Poor lifestyles of Nepalese migrant workers in GCC, Malaysia and India have been widely reported [4, 9, 31, 32]. Studies have suggested that excessive consumption of processed drinks, alcohol and tobacco can be a risk factor for liver diseases, CKD and other cardiometabolic diseases [25, 33, 34].

We found that some did not wear masks while operating in factories. This echoes the issues of poor use of personal protective equipment (PPE) among Nepalese migrants. Some participants even reported developing allergies and becoming habituated to the smell of different chemicals, after working for many years. Since quitting their job due to allergies is not an option for them, their risk-taking behaviours force them to undergo prolonged exposure to occupational pollutants which could pose a risk to kidney function, leading to CKD [35–37].

The majority of our participants indicated possessing health cards issued by their employers; nonetheless, there were discrepancies and inconsistencies regarding their reimbursement of healthcare expenses. Although some ill migrant workers were well treated by employers, some were also sent back to Nepal at their own expense, where they proceeded with further treatment. It is concerning that Nepalese migrant workers are still facing health disparities even after the host countries have introduced migrant-centred reforms in their health systems, as is the case in Qatar [38].

The use of excessive painkillers was commonly reported and there was a belief among participants that medicines/painkillers in the host country do not suit their bodies, hence they prefer bringing them from Nepal (where they are often over the counter without a prescription). There is a risk that migrants might end up consuming excessive doses of medicine without prescriptions. There is an established link between analgesic drugs and kidney diseases [39, 40].

It is mandatory for all aspiring Nepalese migrant workers to complete pre-departure orientation before they can work abroad [1]. However, most participants did not attend the training despite acknowledging its importance as it includes information on maintaining a healthy lifestyle abroad, and on diseases associated with drinking inadequate water. This finding corroborates a previous study with returnee Nepalese migrants from GCC which reported a poor uptake (only half attended) of pre-departure training [32]. In addition, some of our participants who undertook the orientation were not satisfied with the information provided. The effectiveness of this intervention as envisaged is not yet known [41]. Quality issues of the pre-departure training focusing on mental health issues were raised in a study with returnee Nepalese migrants [7]. Further research is necessary to identify barriers to migrants attending the training and evaluate the contents of the training as well as its effective delivery. Surprisingly, none of our participants reported symptoms of kidney diseases before they were diagnosed. It could be due to their low level of awareness of kidney diseases/symptoms or perhaps they may have thought that their systems were related to other health issues. We suggest that incorporating key symptoms of kidney diseases in the pre-departure programme may be beneficial.

## Strengths and limitations

Our study included diverse participants in terms of their age, destination, and occupation, therefore our data are rich. Similarly, most interviews were conducted in the participants'

households which provided a safe and comfortable space to sufficiently express their views. However, all our participants were recruited from *Dhanusha* district of Nepal, returnees from Malaysia or Gulf countries, hence, our study may not capture the views of returnee labour migrants from other districts of Nepal and/or other destination countries. Moreover, we attempted to recruit men and women, but all interviewees were male, as we could not find any female returnee migrants with CKD in the study area.

## Conclusion

Nepalese migrants experienced numerous challenges, such as difficulties accessing drinking water and toilet facilities, compromising their nutritional diet with unhealthy foods, exposure to occupational pollutants, and the use of excessive painkillers, which may elevate the risk of kidney disease. Moreover, disparities in healthcare access and reimbursement systems abroad put them at further risk. There is a necessity for enhanced pre-departure and on-arrival orientation programmes, particularly with an emphasis on kidney health-related topics, to mitigate these risks effectively and promote their general health and well-being.

## Acknowledgments

We would like to thank Green Tara Nepal, our research collaborator, for providing logistic support during data collection in Nepal. We would also like to thank Yagya Adhikari, a PhD student at Bournemouth University, for his support in graphical design.

## Author Contributions

**Conceptualization:** Pramod Regmi, Nirmal Aryal, Arun Sedhain, Radheshyam Krishna K. C., Edwin van Teijlingen.

**Data curation:** Nirmal Aryal, Sankalpa Bhattarai.

**Formal analysis:** Sankalpa Bhattarai.

**Funding acquisition:** Pramod Regmi.

**Investigation:** Sankalpa Bhattarai.

**Methodology:** Pramod Regmi, Nirmal Aryal, Edwin van Teijlingen.

**Project administration:** Sankalpa Bhattarai.

**Supervision:** Edwin van Teijlingen.

**Validation:** Nirmal Aryal, Arun Sedhain.

**Writing – original draft:** Pramod Regmi.

**Writing – review & editing:** Pramod Regmi, Nirmal Aryal, Sankalpa Bhattarai, Arun Sedhain, Radheshyam Krishna K. C., Edwin van Teijlingen.

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
