## [Decision Letter · Decision Letter 0]

17 May 2024

PONE-D-24-14986Exploring lifestyles, work environment and health care experience of Nepalese returnee labour migrants diagnosed with kidney-related problemsPLOS ONE

Dear Dr. Regmi,

Thank you for submitting your manuscript to PLOS ONE. After careful consideration, we feel that it has merit but does not fully meet PLOS ONE’s publication criteria as it currently stands. Therefore, we invite you to submit a revised version of the manuscript that addresses the points raised during the review process.

The required number of reviews have been completed. The reviewer has highlighted several issues with the manuscript, which need to be fixed. Please find their specific feedback in the relevant section at the bottom of this email. Please provide dot by dot response to their comments and make all advised changes (or write rebuttal if necessary).

We look forward to receiving your revised manuscript.

Kind regards,

Ahsan Saleem, PharmD, PhD

Academic Editor

PLOS ONE

Journal Requirements:

2. Please include a complete copy of PLOS’ questionnaire on inclusivity in global research in your revised manuscript. Our policy for research in this area aims to improve transparency in the reporting of research performed outside of researchers’ own country or community. The policy applies to researchers who have travelled to a different country to conduct research, research with Indigenous populations or their lands, and research on cultural artefacts. The questionnaire can also be requested at the journal’s discretion for any other submissions, even if these conditions are not met. 

Please find more information on the policy and a link to download a blank copy of the questionnaire here: https://journals.plos.org/plosone/s/best-practices-in-research-reporting. 

Please upload a completed version of your questionnaire as Supporting Information when you resubmit your manuscript.

"This work was funded by the COLT Foundation, UK."

4. Please note that funding information should not appear in the Acknowledgments section or other areas of your manuscript. We will only publish funding information present in the Funding Statement section of the online submission form. Please remove any funding-related text from the manuscript. 

"The authors declare no conflicts of interest. The funder had no role in the design of the study; in the collection, analyses, or interpretation of data; in the writing of the manuscript; or in the decision to publish the results."

6. In the online submission form you indicate that your data is not available for proprietary reasons and have provided a contact point for accessing this data. Please note that your current contact point is a co-author on this manuscript. According to our Data Policy, the contact point must not be an author on the manuscript and must be an institutional contact, ideally not an individual. Please revise your data statement to a non-author institutional point of contact, such as a data access or ethics committee, and send this to us via return email. Please also include contact information for the third party organization, and please include the full citation of where the data can be found.

7. Your ethics statement should only appear in the Methods section of your manuscript. If your ethics statement is written in any section besides the Methods, please move it to the Methods section and delete it from any other section. Please ensure that your ethics statement is included in your manuscript, as the ethics statement entered into the online submission form will not be published alongside your manuscript. 

8. We note that Figure 1 in your submission contain map images which may be copyrighted. All PLOS content is published under the Creative Commons Attribution License (CC BY 4.0), which means that the manuscript, images, and Supporting Information files will be freely available online, and any third party is permitted to access, download, copy, distribute, and use these materials in any way, even commercially, with proper attribution. For these reasons, we cannot publish previously copyrighted maps or satellite images created using proprietary data, such as Google software (Google Maps, Street View, and Earth). For more information, see our copyright guidelines: http://journals.plos.org/plosone/s/licenses-and-copyright.

1) You may seek permission from the original copyright holder of Figure 1 to publish the content specifically under the CC BY 4.0 license.  

2) If you are unable to obtain permission from the original copyright holder to publish these figures under the CC BY 4.0 license or if the copyright holder’s requirements are incompatible with the CC BY 4.0 license, please either i) remove the figure or ii) supply a replacement figure that complies with the CC BY 4.0 license. Please check copyright information on all replacement figures and update the figure caption with source information. If applicable, please specify in the figure caption text when a figure is similar but not identical to the original image and is therefore for illustrative purposes only.

**Additional Editor Comments:**

 No further comments.

Reviewers' comments:

Reviewer's Responses to Questions

**Comments to the Author**

1. Is the manuscript technically sound, and do the data support the conclusions?

Reviewer #1: Partly

2. Has the statistical analysis been performed appropriately and rigorously? 

Reviewer #1: N/A

3. Have the authors made all data underlying the findings in their manuscript fully available?

Reviewer #1: No

4. Is the manuscript presented in an intelligible fashion and written in standard English?

Reviewer #1: Yes

5. Review Comments to the Author

Reviewer #1: This paper focuses on an interesting return migration health issue. It is overall well written and clear. However, there are quite a few aspects that require revision. I have listed my comments below:

In introduction, it would be worthwhile providing more detail on CKD (defining the health issue, its causes and symptoms, and prevalence in Nepal and destination countries). It would also be good to know more about what policies exist in Nepal on pre-departure and on-arrival orientation programmes.

page 3, line 8. Reference missing.

p.3 lines 13-17 Are these findings from studies taking place in destination countries, or also from source countries? This is unclear with the list of references provided.

p.4, last paragraph of introduction. The link between the knowledge gap this study aims to fill doesnt seem to correlate with the rationale provided in the sentences that precede - which refer more to the lack of diseases surveillance data to know more about prevalence of CKD. However, this study looks more at understanding what risk factors migrants are exposed to when abroad. The study rationale needs to be further clarified.

p4. typo: there is an "s" missing for "interview".

p4. Eligibility criteria: What about asking participants how many years they had been back to Nepal? This seems like key data that is not discussed in this paper.

Figure 1 and table 1 are good illustrations of the sample. However these should be under the findings section, as they present information on the final sample.

In methods, there is no statement as to why women where not included in this study. It is mentioned that women represent only 15% of labour out migration, but it would be important for authors to state more clearly why they decided to not include women in this study.

p.7. Coding of data: How was this done? Was a software used? There is no mention in text, but in supplementary file you mention it was "analysed manually". What does this mean? This information should be in the main manuscript.

Also, how was the analysis done? Thematic analysis is mentioned, but then not elaborated on. This is a very specific analysis method that is often misunderstood. Therefore it is important to describe how it was used.

p.8 It seems the key themes from the thematic analysis are the interview guide headings. This is a decision for the authors, but it does seem there may have been a lack of analysis of the data, and a result that is quite descriptive.

In general throughout the findings, I found the data to be presented in a very descriptive form. Often, I wanted to know more about a statement. For example, why were migrants drinking soft drinks? What did they think of the shared rooms/dorms? Why did they not eat more fruit? Why did they think medications did not suit their body? I am not sure if these questions were not asked in the interviews, or whether its an issue in the analysis stage.

P.9. The first few quotes are very similar and some say the same thing. The second quote doesn't seem to represent the sentence written by the authors.

p.13. Clarify whether the Barber had a medical assessment to conclude his work environment was the cause of his health issues, or whether it was his own perspective? This seems unclear at the moment.

In general, did you ask interviewees about other concurrent health issues? It seems that would have been interesting to factor in, to provide more context to their health.

p.18, first paragraph, lines 3-5. This sentence is unclear. 2nd paragraph: unclear as to why the findings are surprising? This needs more explanation.

In discussion there is generally a lack of literature.

p.19, paragraph 3. "sent back at their own expense". This seems to contradict what is in the findings section - where it states that the company paid for their return to Nepal.

p.19, reference 31. This reference seems to be about Qatar and not other countries. This should be further clarified in text.

p.20, paragraph 1, lines 2-5: I don't think this data was in the findings section.

p.20 Limitation section: This is the first time the district is mentioned by name. This is missing in introduction/methods.

COREQ supplementary file: it seems some of the stated content in this list is not reflected in the paper: item 5, 9, 17, 22 (data saturation), 23, 27.

6. PLOS authors have the option to publish the peer review history of their article (what does this mean?). If published, this will include your full peer review and any attached files.

Reviewer #1: No

---

## [Author Response · Author response to Decision Letter 0]

12 Jun 2024

Editor’s comment

Comment 1. Please ensure that your manuscript meets PLOS ONE's style requirements, including those for file naming. 

Authors’ reply: We have carefully followed the journal requirements. 

Comment 2. Please include a complete copy of PLOS’ questionnaire on inclusivity in global research in your revised manuscript. 

Authors’ reply: Thank you. It is uploaded. 

Comment 3. Thank you for stating the following financial disclosure: "This work was funded by the COLT Foundation, UK."

Authors’ reply: As suggested, we have corrected the statement in the manuscript and have also confirmed this in our cover letter above. 

Comment 4. Please note that funding information should not appear in the Acknowledgments section or other areas of your manuscript. We will only publish funding information present in the Funding Statement section of the online submission form. Please remove any funding-related text from the manuscript. 

Authors’ reply: We confirm that no funding information has appeared in the acknowledgement section or other areas of the manuscript. Thank you.

Comment 5. Thank you for stating the following in your Competing Interests section: 

"The authors declare no conflicts of interest. The funder had no role in the design of the study; in the collection, analyses, or interpretation of data; in the writing of the manuscript; or in the decision to publish the results."

Authors’ reply: We confirm that we have used a correct statement and is also reflected in the cover letter.

Comment 6. In the online submission form you indicate that your data is not available for proprietary reasons and have provided a contact point for accessing this data. Please note that your current contact point is a co-author on this manuscript. According to our Data Policy, the contact point must not be an author on the manuscript and must be an institutional contact, ideally not an individual. Please revise your data statement to a non-author institutional point of contact, such as a data access or ethics committee, and send this to us via return email. Please also include contact information for the third party organization, and please include the full citation of where the data can be found.

Authors’ reply: Thank you. We have provided non-author institutional point of contact. Our data is available from Bournemouth University’s Research Data Repository. Email: bordar@bournemouth.ac.uk

Comment 7. Your ethics statement should only appear in the Methods section of your manuscript. If your ethics statement is written in any section besides the Methods, please move it to the Methods section and delete it from any other section. Please ensure that your ethics statement is included in your manuscript, as the ethics statement entered into the online submission form will not be published alongside your manuscript. 

Authors’ reply: Thank you. We confirm that this information is only available in the method section.

Comment 8. We note that Figure 1 in your submission contain map images which may be copyrighted. All PLOS content is published under the Creative Commons Attribution License (CC BY 4.0), which means that the manuscript, images, and Supporting Information files will be freely available online, and any third party is permitted to access, download, copy, distribute, and use these materials in any way, even commercially, with proper attribution. For these reasons, we cannot publish previously copyrighted maps or satellite images created using proprietary data, such as Google software (Google Maps, Street View, and Earth). For more information, see our copyright guidelines: http://journals.plos.org/plosone/s/licenses-and-copyright.

Authors’ reply: Please note that the map (Figure 1) was created ourselves using the ArcGIS Pro software (Ver 3.1) and the topographic base map was used. There is no copyright issue.

Comment 9. Please include captions for your Supporting Information files at the end of your manuscript, and update any in-text citations to match accordingly. Please see our Supporting Information guidelines for more information: http://journals.plos.org/plosone/s/supporting-information. 

Authors’ reply: Not applicable. 

Reviewer’s Comments to the Author

Comment 1: In introduction, it would be worthwhile providing more detail on CKD (defining the health issue, its causes and symptoms, and prevalence in Nepal and destination countries). It would also be good to know more about what policies exist in Nepal on pre-departure and on-arrival orientation programmes.

Authors’ reply. We have now added paragraphs around CKD, policies and pre-departure/on arrival programmes, for example, we have added:

Chronic Kidney Disease (CKD) is the progressive decline in kidney function for over three months [16]. The global prevalence of CKD is around 10% in the general population, but less common in young adults at around 2% [17, 18]. Diabetes and hypertension are the main risk factors for CKD [17]. A nationally representative survey in Nepal reported a CKD prevalence of 4.4% in 20-to-59-year-olds [19]. A systematic review including all published studies estimated the CKD prevalence of 14.6% in the general population in Nepal, but 18.2% in Malaysia, one of the major destination countries of Nepal’s migrant workers [20]. 

There are policies in place in Nepal, the Government requires mandatory provisions such as pre-departure orientation training and health screening for all aspiring migrants. However, no effective measures are in place to identify, record, improve, and support the health and well-being of returnee Nepali migrants. In some destination countries, for example, GCC and Malaysia, post-arrival health check-ups are required [22].

Comment 2: page 3, line 8. Reference missing.

Authors’ reply: This is corrected. Thank you.

Comment 3: p.3 lines 13-17 Are these findings from studies taking place in destination countries, or also from source countries? This is unclear with the list of references provided.

Authors’ reply: Thank you. This is corrected. 

Comment 4: p.4, last paragraph of introduction. The link between the knowledge gap this study aims to fill doesn’t seem to correlate with the rationale provided in the sentences that precede - which refer more to the lack of diseases surveillance data to know more about prevalence of CKD. However, this study looks more at understanding what risk factors migrants are exposed to when abroad. The study rationale needs to be further clarified.

Authors’ reply: We have re-structured the paragraphs and the last paragraph now reads:

The prevalence of CKD in migrant workers is probably underestimated, however, it is clear that their living and working conditions play a crucial role. Therefore, this qualitative study aims to address our knowledge gap in this field, by exploring possible lifestyle and working environment issues experienced by Nepalese migrants diagnosed with kidney health problems. 

Comment 5: p4. typo: there is an "s" missing for "interview".

Authors’ reply. Corrected. Thank you.

Comment 6: p4. Eligibility criteria: What about asking participants how many years they had been back to Nepal? This seems like key data that is not discussed in this paper.

Authors’ reply: Our interview participants were returnee migrants who returned to Nepal in the past three years with at least two years of work experience in the GCC or Malaysia. We have now added this information. 

Comment 7: Figure 1 and table 1 are good illustrations of the sample. However these should be under the findings section, as they present information on the final sample.

Authors’ reply: Thank you. We have now moved them to the relevant section.

Comment 8: In methods, there is no statement as to why women where not included in this study. It is mentioned that women represent only 15% of labour out migration, but it would be important for authors to state more clearly why they decided to not include women in this study.

Authors’ reply: Our eligibility criteria included both male and female returnee migrants. However we could not find any female participants that meet the criteria (e.g. with CKD). It may be due to the low number of female migrants [most labour migrants (more than 90%) from Nepal are male] and most female work as domestic workers thus may be a low risk for their kidney health. 

Comment 9. Coding of data: How was this done? Was a software used? There is no mention in text, but in supplementary file you mention it was "analysed manually". What does this mean? This information should be in the main manuscript.

Also, how was the analysis done? Thematic analysis is mentioned, but then not elaborated on. This is a very specific analysis method that is often misunderstood. Therefore it is important to describe how it was used.

Authors’ reply: We have added more information around the analysis process. 

Comment 10: p.8 It seems the key themes from the thematic analysis are the interview guide headings. This is a decision for the authors, but it does seem there may have been a lack of analysis of the data, and a result that is quite descriptive.

Authors’ reply: Though our interview guides included topic around the themes we presented, we had had used a semi-structured guides. The themes we presented were generated from the data not according to our interview checklist. We have also revised this to include 7 themes only (Water and toilet issues are now incorporated into the work environment theme.

Comment 11: In general throughout the findings, I found the data to be presented in a very descriptive form. Often, I wanted to know more about a statement. For example, why were migrants drinking soft drinks? What did they think of the shared rooms/dorms? Why did they not eat more fruit? Why did they think medications did not suit their body? I am not sure if these questions were not asked in the interviews, or whether its an issue in the analysis stage.

Authors’ reply: We have added some information/quotes in the finding section. For example:

Some also mentioned problems with facilities, including toilets:

“So for 10 people there was a single kitchen room and toilet. So, sometime you have to wait for a long queue to use toilet.” (IDI #8)

…relied on tap water, perhaps for the first few weeks, supplied in their residences before switching to commercial drinking water due to unpleasant taste. Some avoided drinking tap water due to the fear of getting kidney stones, often buying expensive bottled water:

…Participants suggested that consuming energy drinks or carbonated beverages gave them energy, and that it was widespread among Nepalese migrants:

Most said that they ate green vegetables regularly, eating fruit was seen as expensive and hence rare, for example: 

“ …I used to eat green vegetables…but I rarely ate fruits. I used to eat meat and fish once a week both in the same quantity.” (IDI #7)

Comment 12: P.9. The first few quotes are very similar and some say the same thing. The second quote doesn't seem to represent the sentence written by the authors.

Authors’ reply: We have revised this. Thank you. 

Comment 13: p.13. Clarify whether the Barber had a medical assessment to conclude his work environment was the cause of his health issues, or whether it was his own perspective? This seems unclear at the moment.

Authors’ reply. Thank you. We have revised this. This now reads as: 

Another participant, who was employed as a barber and suffered from severe kidney infections, felt there was a link between CKD and having been exposed to various chemicals in the workplace, including lotions, highlighters, straightening creams, fragrances, and cosmetics with unpleasant odours

Comment 14: In general, did you ask interviewees about other concurrent health issues? It seems that would have been interesting to factor in, to provide more context to their health.

Authors’ reply: We focused mainly around the CKD however we had explored their health before going abroad. None reported any health issues. 

Comment 15: p.18, first paragraph, lines 3-5. This sentence is unclear. 2nd paragraph: unclear as to why the findings are surprising? This needs more explanation.

Authors’ reply: We have revisited this. Thank you.

Comment 16: In discussion there is generally a lack of literature.

Authors’ reply: We have added additional literature in the introduction, method and discussion but some of four findings are novel. There are no previous studies/literature to link with them. 

Comment 17: p.19, paragraph 3. "sent back at their own expense". This seems to contradict what is in the findings section - where it states that the company paid for their return to Nepal.

Authors’ reply: We have corrected this. This now reads as:

Although some ill migrant workers were well treated by employers, some were also sent back to Nepal at their own expense, where they proceeded with further treatment…

Comment 18: p.19, reference 31. This reference seems to be about Qatar and not other countries. This should be further clarified in text.

Authors’ reply: Thank you. Corrected.

Comment 19: p.20 Limitation section: This is the first time the district is mentioned by name. This is missing in introduction/methods.

Authors’ reply: Now added. Thank you.

Comment 20: COREQ supplementary file: it seems some of the stated content in this list is not reflected in the paper: item 5, 9, 17, 22 (data saturation), 23, 27.

Authors’ reply:

---

## [Decision Letter · Decision Letter 1]

5 Jul 2024

PONE-D-24-14986R1Exploring lifestyles, work environment and health care experience of Nepalese returnee labour migrants diagnosed with kidney-related problemsPLOS ONE

Dear Dr. Regmi,

Thank you for submitting your manuscript to PLOS ONE. After careful consideration, we feel that it has merit but does not fully meet PLOS ONE’s publication criteria as it currently stands. Therefore, we invite you to submit a revised version of the manuscript that addresses the points raised during the review process.

Kind regards,

Ahsan Saleem, PhD

Academic Editor

PLOS ONE

Journal Requirements:

Additional Editor Comments (if provided):

Reviewers' comments:

Reviewer's Responses to Questions

**Comments to the Author**

1. If the authors have adequately addressed your comments raised in a previous round of review and you feel that this manuscript is now acceptable for publication, you may indicate that here to bypass the “Comments to the Author” section, enter your conflict of interest statement in the “Confidential to Editor” section, and submit your "Accept" recommendation.

Reviewer #1: (No Response)

2. Is the manuscript technically sound, and do the data support the conclusions?

Reviewer #1: Yes

3. Has the statistical analysis been performed appropriately and rigorously? 

Reviewer #1: N/A

4. Have the authors made all data underlying the findings in their manuscript fully available?

Reviewer #1: Yes

5. Is the manuscript presented in an intelligible fashion and written in standard English?

Reviewer #1: Yes

6. Review Comments to the Author

Reviewer #1: Thank you for the responses to my comments. The revisions have improved the paper.

I have two remaining comments from the first review round:

Comment 8: Description of gender of sample. Thank you for adding a comment on women's participation in the study in the limitations section. However, I notice there is no mention of gender of participants on page 6 (either in text or table). It would be important to note gender of sample.

Comment 20: I did not see a response to my comment on the COREQ supplementary file. My original question was: COREQ supplementary file: it seems some of the stated content in this

list is not reflected in the paper: item 5, 9, 17, 22 (data saturation), 23, 27. Please clarify.

7. PLOS authors have the option to publish the peer review history of their article (what does this mean?). If published, this will include your full peer review and any attached files.

Reviewer #1: No

---

## [Author Response · Author response to Decision Letter 1]

12 Jul 2024

Dear Editor

Thank you for sharing the further suggestions by the reviewers. We have addressed each of the comments below (in red fonts). Both clean and track change versions of the manuscript are attached for your kind perusal. 

We look forward to hearing your reply soon.

Yours sincerely,

Pramod Regmi, on behalf of co-authors

Comment 8: Description of gender of sample. Thank you for adding a comment on women's participation in the study in the limitations section. However, I notice there is no mention of gender of participants on page 6 (either in text or table). It would be important to note gender of sample.

Authors reply: Thank you for the suggestions. We have added the gender characteristics of participants in the findings section. Page number 6, the last paragraph reads ‘All participants in this study were male returnee migrants, aged 24 to 45….’

Comment 20: I did not see a response to my comment on the COREQ supplementary file. My original question was: COREQ supplementary file: it seems some of the stated content in this

list is not reflected in the paper: item 5, 9, 17, 22 (data saturation), 23, 27. Please clarify.

Authors reply: Thank you. We have addressed each of your comments below.

COREQ item 5. What experience or training did the researcher have:

Authors reply: This information is available on page 5 of the manuscript under the ‘Interview tool, approach and setting’ heading. We have the following information there: “All interviews were conducted by SB, an experienced male qualitative researcher with a Master’s degree in Global Health. PR and NA oriented the researcher about various aspects of qualitative data collection and ethical considerations”

COREQ item 9-What methodological orientation was stated to underpin the study? e.g. grounded theory, discourse analysis, ethnography, phenomenology, content analysis

Authors reply: Our study is based on the Generic Qualitative Approach. We have added this with a reference. Please see the opening sentence of the last paragraph on page 4.

COREQ item 17. Were questions, prompts, guides provided by the authors? Was it pilot tested?

Authors reply: This information is available within the Interview tool, approach and setting section (page 5). For example, related sentences are:

…Our interview guide included topics such as living and lifestyles abroad, work environment, CKD experience while working abroad, access to health, cost related to kidney health problems treatment abroad, and pre-departure and on-arrival orientation experience.

… A concurrent data organisation and analysis approach was followed to enable the interviewer to focus on any emerging issues in subsequent interviews.

COREQ item 22. Was data saturation discussed?

Authors reply: This information is available within the Interview tool, approach and setting section (page 5). This reads “…Interviews were conducted up to a saturation point where no new information were provided by the participants”

COREQ item 23. Were transcripts returned to participants for comment and/or correction?

Authors reply: This information is available on page 6 which reads:

…interview transcripts were not returned to the participants for comments, a concept known as member checking, since they were reluctant to read.

COREQ item 27. What software, if applicable, was used to manage the data?

Authors reply: This information is available within the Data management and analysis section. This reads:

The data analyses followed the six steps outlined by Braun and Clarke [27], and Microsoft Excel was used to organise, code and track themes in the data.

---

## [Decision Letter · Decision Letter 2]

8 Aug 2024

Exploring lifestyles, work environment and health care experience of Nepalese returnee labour migrants diagnosed with kidney-related problems

PONE-D-24-14986R2

Dear Dr. Regmi,

We’re pleased to inform you that your manuscript has been judged scientifically suitable for publication and will be formally accepted for publication once it meets all outstanding technical requirements.

Kind regards,

Rabie Adel El Arab

Academic Editor

PLOS ONE

Additional Editor Comments (optional):

Reviewers' comments:

Reviewer's Responses to Questions

**Comments to the Author**

1. If the authors have adequately addressed your comments raised in a previous round of review and you feel that this manuscript is now acceptable for publication, you may indicate that here to bypass the “Comments to the Author” section, enter your conflict of interest statement in the “Confidential to Editor” section, and submit your "Accept" recommendation.

Reviewer #2: All comments have been addressed

Reviewer #3: All comments have been addressed

2. Is the manuscript technically sound, and do the data support the conclusions?

Reviewer #2: Yes

Reviewer #3: Yes

3. Has the statistical analysis been performed appropriately and rigorously? 

Reviewer #2: Yes

Reviewer #3: N/A

4. Have the authors made all data underlying the findings in their manuscript fully available?

Reviewer #2: No

Reviewer #3: Yes

5. Is the manuscript presented in an intelligible fashion and written in standard English?

Reviewer #2: Yes

Reviewer #3: Yes

6. Review Comments to the Author

Reviewer #2: I am happy with the revised version of the paper. I like the research question, the structure of the paper, the quality of writing, and the way the authors describe their empirical proceeding and results. Most importantly, the authors have addressed all the issues stated in my referee report for the first version appropriately.

Reviewer #3: An insightful and intriguing article that raises a serious concern: the poor migrant labour population is subjected to conditions that predispose them to devastating kidney illnesses. It is distressing that exploitation continues to occur in places where South Asians work, such as Qatar and Malaysia. To go deeper into this topic, I recommend that the authors perform follow-up research with social workers, social policy advocates, public health professionals, and human rights activists (both local and international). It is critical to employ welfare professionals to conduct a full assessment of this sector.

While pre-immigration concerns are crucial, importing countries must also be accountable for the well-being of its workforce. Many of these people live abroad in relative safety- bubbles for a while, frequently remitting monies to their family, only to return to their native nations as patients. Addressing such complicated issues necessitates a multidisciplinary approach to ensure a dignified future for all individuals concerned.

I support the publication of this article.

Prof Venkat Rao Pulla - Brisbane, Australia

7. PLOS authors have the option to publish the peer review history of their article (what does this mean?). If published, this will include your full peer review and any attached files.

Reviewer #2: No

Reviewer #3: **Yes: **Prof Venkat Rao Pulla, Adjunct Associate Professor, Department of Social Work, James cook University, Australia

Distinguished Fellow, FAASW

Foundation Professor of Strengths-Based- Social Work Practice

Brisbane Institute of Strengths-Based Practice

Inaugural Fellow, Australian College of Researchers

& Life Member, Australian Institute of International Affairs

Editor-in-Chief (joint), Journal Space and Culture, India ( In Scopus)

Associate Editor, SN Social Sciences- Open Springer

Editorial Board: The Journal of Applied Research and Innovation (JARI)

Editorial Board: The International Journal of Innovation, Creativity and Change ( In ERA)

https://orcid.org/0000-0003-0395-9973

---

## [Editor Report · Acceptance letter]

12 Aug 2024

PONE-D-24-14986R2 

PLOS ONE

Dear Dr. Regmi, 

I'm pleased to inform you that your manuscript has been deemed suitable for publication in PLOS ONE. Congratulations! Your manuscript is now being handed over to our production team.

Kind regards, 

on behalf of

Dr. Rabie Adel El Arab 

Academic Editor

PLOS ONE